# Environmental Enrichment for Rainbow Trout Fingerlings: A Case Study Using Shelters in an Organic Trout Farm

**DOI:** 10.3390/ani13020268

**Published:** 2023-01-12

**Authors:** Julia Eidsmo, Lone Madsen, Lars-Flemming Pedersen, Alfred Jokumsen, Manuel Gesto

**Affiliations:** 1Section for Aquaculture, The North Sea Research Centre, DTU Aqua, Technical University of Denmark, Willemoesvej 2, 9850 Hirtshals, Denmark; 2Unit for Fish and Shellfish Diseases, Public Sector Consultancy, DTU Aqua, Technical University of Denmark, Kemitorvet Building 202, 2800 Kongens Lyngby, Denmark

**Keywords:** enrichment, fish, aquaculture, welfare, stress, shelter

## Abstract

**Simple Summary:**

Enriching the places were captive fish live, i.e., tanks, ponds or cages, has demonstrated potential to improve the wellbeing of the fish, which is a matter of increasing concern in the aquaculture sector. Enriching strategies are diverse but studies on the feasibility of their implementation in real farming scenarios are scarce. Here, the feasibility of using structural enrichment in the form of plastic shelters to improve the welfare of rainbow trout was studied in an organic fish farm. It was demonstrated that the use of simple plastic shelters is technically feasible, since the shelters induced little extra work in the farm routines and had no negative effects on fish performance, health or mortality. However, different to laboratory-based studies, the fish did not develop a clear shelter-seeking behavioral response when disturbed. This could be related to the short duration of the study, and therefore, it is recommended that enrichment in real scenarios should be tested covering a relevant part of the life cycle of the fish.

**Abstract:**

Physical enrichment can improve the welfare of captive fish. Previous research has shown that fish often show preference for enriched environments, which can also result in improvements in growth performance. However, effects of enrichment are not always positive and the design and extent of the enrichment needs to be carefully considered. In this regard, information in real aquaculture scenarios is limited. The aim of this study was to serve as a proof of concept to test the feasibility of using simple PVC immersed shelters as a tool for better welfare in an organic rainbow trout farm. Our shelters induced little extra work in farm routines and had no negative effects on fish performance, health or mortality. The behavioral assessment pointed to a preference for sheltered areas in undisturbed conditions. However, no benefits were observed in terms of stress responses during standardized stress tests, and fish showed no obvious shelter-seeking behavior after disturbance. The results in terms of shelter-seeking behavior were probably limited by the short duration of the experiment, which was due to the farm’s routines and needs. It is recommended that strategies for enrichment in real scenarios should be tested covering a relevant part of the life cycle of the fish in captivity, to fully account for their potential to improve welfare in aquaculture.

## 1. Introduction

Traditional hatcheries often provide rearing environments that differ substantially from the natural habitat of salmonids. This could be one of the reasons behind the high mortalities of captive-reared salmonids when released in the wild [1,2]. The use of environmental enrichment (EE) to increase the complexity of the rearing environment can improve the welfare of captive fish, for example by decreasing aggression, and stress- or fear-related behaviors [3,4,5]. Among different types of EE, the use of physical enrichment, by adding different structures to the rearing units (substrates and other physical elements including shelters) has gained interest in the recent past [5,6,7]. From the results of previous studies on the topic, it is now clear that, in spite of the huge potential of physical enrichment for improving the welfare of captive fish, there is a need to carefully plan the design, scale and timing of the application of the enrichment; this because suboptimal enrichment strategies can result in neutral or even detrimental effects for the fish, such as increased aggression or reduced growth and condition [8].

Another important concern about current research on EE for captive fish is the fact that most available studies were performed in small scale rearing units or in controlled experimental conditions that do not represent the conditions that farmed fish are exposed to in real aquaculture. Available information about the effects of physical enrichment strategies in real scenarios, with limited control of the environment within or surrounding the rearing units (variable fish stocking densities, light and sound environments, variable water quality, etc.), is still scarce. An exception might be the pioneer research by Barnes and colleagues [7,9,10] using vertically suspended structures in salmonid hatcheries [7,9,10]. Nevertheless, the general lack of specific information about the applicability of different types of physical enrichment likely explains the very limited incorporation of enrichment strategies in current farming practice. In their recent review concerning EE in aquaculture, Ref. [6] highlighted the need to perform studies for enrichment optimization at the commercial scale.

Previous studies showed that overhead covers providing shade and visual isolation could be beneficial to fish performance in salmonids [11,12]. We hypothesized that immersed covers could provide the same benefits, and at the same time provide better shelter and a higher environmental complexity to the reared fish. Therefore, this case study aimed at assessing the feasibility of implementing the use of physical enrichment as shelters consisting of immersed plastic covers (made of polyvinyl chloride, PVC) in a real farming scenario, an organic rainbow trout farm in Denmark. We aimed to investigate the effects of the shelters in relation to fish growth performance, health and welfare, and to test the practicalities involved in the use of shelters during normal farming operation. The covers used in the study were PVC plates designed to ease installation/removal and cleaning procedures. The screen shelters were similar to those previously tested at a smaller scale in the laboratory, which the fish used as refuge when disturbed [13]. The growth performance (tank-based biomass gain and feed conversion ratio, individual mass and condition factor) and the external appearance of the fish (injuries to fins, skin, mouth and operculum) were assessed. Additionally, the overall behavior of the fish was compared and evaluated on three occasions (after two, three and four weeks from the start of the experiment), based on the distribution patterns in the tanks with and without cover. Two acute stress challenge trials (two and four weeks after the start) were also performed, using the gill levels of cortisol as stress marker, to assess potential effects of the shelters under threatening situations for the fish.

## 2. Materials and Methods

### 2.1. Ethical Statement

The care and use of animals complied with Danish and EU animal welfare laws. The use of fish in this study was approved by the Animal Experiments Inspectorate from the Ministry of Environment and Food of Denmark, under the license number 2019-15-0201-00330.

### 2.2. Fish

The experiments were carried out at an organic rainbow trout (*Oncorhynchus mykiss*) farm (Ådal dambrug, Vejle, Denmark). Eyed eggs were received in May 2021 and hatched at the facility. At first feeding, the fry were moved to circular fiberglass tanks (2 m diameter, 0.45 m water height, 1414 L), part of a roofed flow-through system, and were reared for approximately three months before the start of the trials. Just before the start of the trials (when fish were around 3 g on average), the fish were graded and vaccinated against *Yersinia ruckeri* (immersion vaccine AquaVac^®^ ERM vet., Intervet International B.V, Boxmeer, Holland). 

### 2.3. Experimental Setup

The experiments were performed in the autumn, directly in the same farm facility where fish were reared, and following the usual protocols of the farm. The setup consisted of six circular fiberglass tanks. Three tanks were fitted with a shelter (shelter treatment) and three were left barren (control treatment). After grading and vaccination, the experiment was started by allocating 5 kg (measured to nearest 100 g) of fry into each of the six tanks from a common pool. Three batches of 500 g of fish were collected and the number of individuals in each batch was counted to estimate the starting individual mass, which was 3.18 g (SEM = 0.17 g). The initial stocking density was 3.54 kg m^−3^, and fish numbers were estimated at 1556–1588 fish per tank. The experiment ended 30 days after start, when the fish were captured for grading and distributed to new tanks in the facility according to their size.

Shelters consisted of immersed PVC covers, covering one third of the tank surface (to provide enough space for the fish while facilitating the access to light and food and the tank-cleaning procedures) and were built in dark gray, 5 mm thick high-density PVC, in a circular-section shape so they could be allocated against the tank wall (Figure 1). The shelters were placed on eight legs made of PVC pipes (30 cm high), spanning two thirds of the water column. Shelters were placed in the opposite side of the tanks to the feeders, to avoid feed collection on top. All six experimental tanks formed a row and every second tank was fitted with a shelter.

During the experimental period, high-quality well water was delivered to the fish tanks at a rate of 100 L min^−1^. Water quality parameters were measured daily (temperature and oxygen saturation—OxyGuard Handy Polaris TGP, OxyGuard International A/S, Farum, Denmark) or weekly (pH—HQ40D digital multimeter, Hach, Ames, ID, USA; ammonia-N, nitrite-N and nitrate-N—MQuant^®^ 1.11117, 1.10057, 1.10020, Merck, Darmstadt, Germany). Temperature remained almost constant at 9 °C (SD = 1 °C) and oxygen saturation was within 85% to 90%. pH was 7–7.5 and total ammonia-N, nitrite-N and nitrate N, was always below 0.4 mg L^−1^, 0.15 mg L^−1^ and 2.3 mg L^−1^, respectively. The tanks were placed in a hatchery facility without temperature control, illuminated by both ambient natural light and fluorescent lamps that were left on 24 h/day. Light intensity was measured at mid-day with a handheld light meter (Amprobe LM-120, Beha-Amprobe GmbH, Glottertal, Germany) and ranged between 44 lux and 92 lux, when measured directly above the water surface. Light-intensity under shelters (measured in empty tank) was 0.5 lux. Day length varied between approximately 13 h at experimental start and 10.5 h at the end.

Tanks were left undisturbed and were only accessed during the daily cleaning routine (less than 5 min per tank). This entailed lowering the water level 10 cm for a short period and scrubbing the tank bottom and collecting dead fish. Shelters were moved briefly for daily cleaning and moved back to the original placement.

The rainbow trout fry were fed organic granulated feed of 0.9–1.6 mm (54% crude protein, 15% crude fat, Aller Aqua, Christiansfeld, Denmark). Feeding rations were based on [14], with an expected feed conversion ratio of 0.6 and daily feed rations of 3% of fish mass day^−1^, adjusted weekly. Feed was dispensed by automatic feeders (Linn Gerätebau GmbH, Lennestadt, Germany) that were electronically programmed to release small amounts of feed at regular intervals, 24 h a day. Actual feed fed over the experimental period was calculated by weighing the feed put into feeders at start (measured to nearest gram), any additional replenishing and subtracting feed leftover at end of study. Each tank received 3474 g (SD = 127 g) feed during the period.

### 2.4. Tank-Based Biomass, FCR and Mortality

Total tank biomass was measured (to nearest 100 g) at the start and end of the study. FCR for each tank was calculated at the end of the study period, based on total tank biomass weight gain and amount of feed fed (measured to nearest g) per tank. Mortality was monitored daily in all tanks. For mortality assessment, initial fish numbers were assumed to be equal among all tanks (as of 1572 fish per tank), and the fish sampled for size and for stress trials were not considered in the calculations.

### 2.5. Individual Performance and External Health

A number of individuals (*n* = 20 at start and *n* = 12 on all other dates) were collected randomly from each tank every week to assess individual weight, fork length and condition factor. Fish were handled under lethal anesthesia (140 mg L^−1^ benzocaine solution, Sigma-Aldrich, Steinheim, Germany). No individual measurements were taken after the first week due to technical failure of the weighing equipment carried to the farm. Fulton’s condition factor K was calculated as: K = 100 × Weight (g) × Length (cm)^−3^ [15]. External health was assessed by visual inspection of the fish prior to, midway and at the end of study. Fish were examined for lesions on skin, fins (pectoral, dorsal and caudal), and eyes, based on visual scoring systems [16].

### 2.6. Stress Trials

Fish in the experimental tanks were submitted to two acute stress-challenge trials, one at two weeks after the start of the experiment (fish around 4 g of mass on average) and another one at the end of the experimental period, four weeks after the start (fish around 6 g of mass on average). The standardized stressor was the same in both occasions and consisted in chasing by moving a small net inside the water for 2 min. Three fish per tank were sampled at 0 min (pre-stress control) and at 15 min, 50 min and 90 min after the start of the 2 min-stress protocol. Sampled fish were lethally anesthetized in a 140 mg L^−1^ benzocaine solution. Fish were then weighed and the gill arches on the left side of the fish were collected and kept on dry ice. At the end of the day, all samples were stored in a −80 °C freezer until analysis. Cortisol was analyzed in the fish gill tissue using a cortisol ELISA kit (Ref: 402710, Neogen Europe, Ayr, UK). Gill tissue was processed as previously described [17]. Briefly, gill tissue was homogenized in phosphate-buffered saline (pH = 7.50). An aliquot of the homogenate was used without dilution in the ELISA assay. A second aliquot was assayed for protein using the bicinchoninic acid method [18]. Tissue protein levels were used to normalize the cortisol data.

### 2.7. Use of Shelters and Fish Behavior after Disturbance

In resting conditions, fish in the tanks tended to swim calmly against the circular current, well dispersed over the whole tank area. Preliminary observations of the fish showed that the presence of the shelter did not seem to modify the swimming activity of the fish, but might modify their dispersion between sheltered and not sheltered areas. Therefore, the use of shelters in undisturbed conditions was estimated from zenithal pictures of the fish tanks. The percentage of the tank-exposed areas (not sheltered) occupied by fish was quantified in both control and shelter tanks and compared. Using this approach, a lower percentage of coverage by fish (with respect to control tanks) in the exposed areas of sheltered tanks would imply a higher presence of fish under the shelters and vice-versa. Zenithal pictures from each tank were obtained in six occasions, during the morning and afternoon on week 2, 3 and 4 from the start of the experiment.

A protocol was also developed to investigate the shelter seeking behavior of the fish upon external disturbance. A knocking stressor was therefore used (on the same six occasions when the pictures described before were taken), and new “post-disturbance” zenithal pictures were taken from each tank. The knocking stressor consisted of knocking the plastic head of a broom against the outside wall of each tank for five seconds. The knocking was done opposite to the water inlet of the tank and close to the extreme of the shelter, when present, to minimize the movements of the fish towards or away from it. Unfortunately, the zenithal pictures after disturbance could not be used to quantify shelter use as described above. This because fish tended to group after the knocking stressor, many fish shading others in the zenithal pictures, making the zenithal area coverage inadequate for accounting for shelter use. Therefore, the pictures were used for a qualitative description of the fish behavior upon disturbance.

### 2.8. Picture Collection and Image Analysis

Digital images (12MP) were obtained with an action camera (GoPro Hero 8, GoPro Gmbh, Munich, Germany) on a modified tripod. The camera was placed above the water surface of each tank, one meter from the tank bottom, in order to capture the entire tank in one image. Images from each tank were obtained 5 min after the placement of the camera-rig above the tank, as this was estimated to be adequate time for the fish to habituate to its presence. A series of images for each tank, in resting and post-disturbance conditions, was taken twice (morning and afternoon of the same day), on week 2, 3 and 4 after the start of the experiment. Five images were taken in resting conditions (5 frames s^−1^). Five pictures were obtained immediately after the knocking stressor (1 frame s^−1^). The framerate was lower for the second image subset to better capture the fish’s immediate reaction upon disturbance. Approximately 5 min passed before the image series was repeated for the next tank, to let the fish in the neighboring tank “settle down” after the noise inflicted by the knocking stressor. The camera was operated remotely to avoid interference due to observer presence.

In the images taken in resting conditions, the contrast of the fish relative dark color against the lighter tank bottom was utilized in a method of automatic image analysis, inspired by [19,20]. Images were converted into binary images by processing in Fiji/Image J software (https://fiji.sc accessed on 1 December 2021), in the following steps: (I) The images from each tank were changed from RBG (color) to 8-bit gray scale. (II) Images were cropped to leave only the area of exposed tank bottom. In control tanks, this was the entire tank bottom, and in the shelter treatment, the two-thirds of the tank bottom not covered by the shelter. Images were inverted. (III) Image thresholding through a series of steps; background was subtracted (rolling ball, radius 50 pixels) and histogram normalized (saturated pixels = 0.3%) before contrast was increased (min. and max. pixel intensity value of 50/255). The result was a binary image where the fish were represented as black pixels on a white background. The image analysis process is illustrated in Appendix A. The software was then used to calculate the percentage of black pixels present in the area corresponding to the exposed tank bottom. This was expressed as the “Bottom area coverage” (BAC) and was used as an inverse measure of the extent of shelter use in the sheltered tanks. For example, BAC = 10% meant that only 10% of the exposed tank bottom was covered by fish and indicated that more fish were dispersed under the shelter, compared to a BAC of 20%. The BAC for the five pictures were averaged. Data consisted of 18 BAC values per shelter group (six sampling events, three replicated tanks).

### 2.9. Pathogen Diagnosis: Sampling and Analyses

Ten days into the experiment, the attending veterinarian diagnosed an outbreak of Rainbow Trout Fry Syndrome (RTFS) as well as unspecified gill disease based on macroscopic observations on the farm, also affecting the fish that were part of the experiment. The affected fish showed darker skin color, pale gills, anemia and signs of bacteremia and sepsis. Treatment was administered based on suspicion of *Flavobacterium psychrophilum* being the causative agent of RTFS. Treatment consisted of water disinfection (Aqua Oxides 15%—SC Sørensen, Thisted, Denmark, two times daily for two days) and antibiotics (florfenicol—Norfenicol^®^, Norbrook, Newry, UK) daily administered in the feed for 10 days. Two weeks after the end of the antibiotic treatment, groups of fish (5 fish per tank, *n* = 15 per shelter group) were killed by immersion in lethal anesthesia (140 mg L^−1^ benzocaine) and sampled for bacteriology to assess potential differences in pathogen loads between groups. Fish were sampled randomly, and thereafter transported to the bacteriology laboratory on ice. Samples were taken from kidney and gills by sterile loops and streaked onto blood agar and TYES (tryptone yeast extract salts) agar [21]. Plates were incubated at 15 °C for up to 10 days and read at regular intervals during this period. Colonies on TYES agar plates resembling *Flavobacterium psychrophilum* were confirmed or rejected by using MALDI-TOF [22].

### 2.10. Statistical Analysis

All analyses were carried out in SigmaPlot (v.14.0. Systat Software Inc., San Jose, CA, USA). Tank-based final biomass, gained biomass, used feed and FCR were tested with Student’s *t*-tests. Individual mass and Fulton’s K were averaged per tank, and the effects of time (week of study) and treatment (Control vs. Shelter) were tested with two-way repeated measures ANOVA. Mortality was assessed with Kaplan-Meier curves, using the log-rank test. First, mortality curves for each tank were compared within treatments (Control or Shelter). Since no differences were found among replicated tanks, mortality data was merged and compared between treatment groups (Control vs. Shelter). The effects of treatment and time post-stress on gill cortisol levels was assessed by two-way ANOVA. The behavioral data about the area covered by fish in the obtained pictures was assessed by two-way repeated measures ANOVA, using treatment (Control vs. Shelter) and event (six different assessment events) as factors. In every case, significance level was set up at *p* < 0.05.

## 3. Results and Discussion

Different physical structures can be used to increase the complexity of the rearing units, thus providing enriched environments to captive fish. The use of structurally richer environments has been shown to have potential benefits to fish welfare, given that the extent of the enrichment and the timing for its application is adequate for the species [5,6,8]. However, the large majority of studies about enrichment have been performed in controlled conditions, most often in a laboratory environment. There are pronounced differences in the conditions at real farms compared to research facilities, and therefore, we tried in this study to assess the potential for using simple shelters in a real farming scenario in relation to facilities and farming operational routines. Barnes and colleagues tested the application of other types of structural enrichment (vertically suspended structures) in hatchery-like conditions in different salmonids. They found that the structures could be applied without relevant problems for the routine operations in the hatchery and the effects on fish performance were generally positive [7,23].

### 3.1. Growth Performance, Health and Mortality

The simple shelters used here were designed to enrich the tank environment by providing a place for the fish to hide, but also to be simple to move/remove to facilitate daily tank-cleaning procedures. The covers were supported by eight thin legs to limit the disturbance to the water circulation and self-cleaning properties of the circular tanks [24]. The results showed that the shelters had no effects on the growth performance of the trout juveniles during normal farm operation. Tank-based biomass gain and FCR did not differ between Control and Shelter tanks at the end of the experiment (Figure 2). Individual weight increased every week in a similar way in both groups (Figure 3a). Condition factor was neither affected by time or the shelters (Figure 3b). In general, external health of the animals was very good, with no incidence of fin, skin or mouth/snout lesions in the fish. During the observations along the experiment, the only observed external lesion was a shortening of the operculum, which affected 10%–15% of the observed individuals (similar prevalence in both treatment groups), generally with lesions of low severity. The etiology of opercular damage is poorly known [16], but opercula have an important role in respiratory mechanics in fish and shortened opercula can reduce the ability to pump water through the gills, reducing oxygenation efficiency and exposing the delicate tissue to mechanical damage and pathogens [16,25,26].

The mortality during the experiment was low, between 3.5% and 4.5% for all experimental tanks. Mortalities increased at the middle of the experimental period due to the RTFS outbreak, mostly between days 10 and 22 of the experiment (Figure 4). There were no differences in mortality between replicates for any of the groups, or between the treatments (Control vs. Shelter). All in all, growth performance, external appearance and mortality data showed that shelters did not induce any negative effect when compared to the control tanks. In this regard, previous studies had shown positive [4,27], neutral [13] and negative effects [28] on growth performance. This clearly emphasizes that the design, extent and timing for application of covers or other means of enrichment should be carefully considered when aiming to support fish welfare [5,6,8]. The bacteriological analysis after the end of the suspected *Flavobacterium psychrophilum* outbreak showed that samples taken from the kidney were sterile in all fish, indicating that the antibiotic treatment was effective. The gill samples were not found to be sterile. One fish of the fifteen tested in the Control group was positive for *F. psychrophilum*, whereas five fish of fifteen resulted in positive in the Shelter group. This indicated that either the pathogen survived in the surface of the fish after both water treatment and dietary antibiotic administration, or the body surface was recolonized by *F. psychrophilum*. The amount of data is too limited to draw any conclusion in this regard when it comes to potential differences between the two groups, but the potential effects of additional structures within the rearing tanks on the presence and transmission of pathogens should be considered in future studies.

### 3.2. Stress-Challenge Trials

Two stress-challenge trials were performed during the experimental period, one at midterm and one at the end. These trials aimed at testing whether the shelters provided protection against acute stress and whether the effects of the shelters depended on the time the fish were given to adapt to them. In both trials, the stress marker cortisol increased in fish gill tissue after the net chasing protocol, validating the challenge trials, but the results showed that the stress levels of the fish were similar for both Control and Shelter groups (Figure 5), pointing to a lack of effect of shelters on the fish response to acute chasing stress. In spite of the lack of differences among treatment groups, the kinetics of the cortisol response were somewhat different in both trials. At midterm (week 2), cortisol was clearly elevated at 15 min after the stress but recovered fast and cortisol levels at 50 min were already not significantly different than pre-stress levels. At the end of the experiment (week 4), the same stressor resulted in a fast elevation of gill cortisol at 15 min, but cortisol remained elevated throughout the trial with no clear signs of recovery at 90 min post-stress (Figure 5). The reason behind this difference in the stress response is unclear, given that different factors are known to affect the stress response of the fish, including age and size, environmental factors or previous experience [29,30]. The developmental stage and size of the fish (which were 50% larger at week 4 when compared to week 2, from 4.1 to 6.2 g on average) could have affected the cortisol kinetics. In this regard, it is known that cortisol tends to peak in the blood of trout after longer times in larger fish [31]. Another potential cause of the differences in the cortisol response could be related to the previous events experienced by the fish, since at week 4, fish had already been exposed to the first stress challenge trial, to several knocking protocols for behavioral assessment, and gone through a disease outbreak. These episodes could have resulted in an allostatic load that might have affected subsequent responses to stress [32].

### 3.3. Behavioral Observations

In any case, even in the absence of any positive effect on growth performance or on the responses to acute stress, the covers were preferred by the fish, since horizontal distribution was favored towards the sheltered areas according to our behavioral assessment. The numbers of individuals in the sheltered/exposed areas in resting conditions could not be determined directly due to their small size and the high number of individuals in the tanks. The indirect approach based on fish coverage of the exposed areas pointed to a higher presence of fish in the sheltered than in the non-sheltered areas. Specifically, in our zenithal pictures fish in control tanks covered an average of 17.1% of the exposed area, but this percentage was significantly reduced to a 10.8% of the exposed area in sheltered tanks (Figure 6), pointing to higher relative presence of fish under the sheltered area. This indicates that fish showed preference for the sheltered area, and therefore, the presence of the shelters is suggested to have a positive welfare effect from a feelings-based welfare point of view [4,33,34].

The assessment of the dispersion of the fish by zenithal photographs was used in resting conditions, in which the fish tended to swim against the water current, well distributed across the available surface (Figure 7). However, the same method could not be applied to quantify any tendency regarding potential shelter-seeking behavior during our disturbance protocol. Fish responded to the knocking grouping together, away from the tank walls, making zenithal pictures futile to assess the use of the shelters. According to our observations, this behavior was similar in both control and sheltered tanks. Even when the qualitative assessment did not allow assessing subtle differences in fish behavior, it was clear that the fish did not respond with immediate shelter-seeking behavior, in contrast to observations in a previous study in a laboratory environment [13]. The lack of shelter-seeking behavior could be related to the limited duration of the experiment, and therefore, limited adaptation time for the fish. In our previous study, involving larger fish, it took several weeks for the fish to develop and strengthen that behavioral pattern. Longer studies, comprising a longer part of the fish development in the farms will be needed to clarify this and to understand the full potential of the use of shelters in trout aquaculture. Furthermore, our approach was also limited in the sense that it focused only on the most immediate reaction (a few seconds) of the fish to the disturbance; it would be advantageous in future studies to assess the behavior of the fish for a longer time after disturbance, to better understand the potential advantages of the shelters during acute stress.

### 3.4. Conclusions

In summary, the fish showed preference for the shelters and tended to swim more under sheltered areas when available. However, the shelters provided no observable protection against a chasing stressor, likely because the fish did not develop a clear shelter-seeking behavioral response to disturbance in the period of study. The experiment was carried out in a real farming scenario, and fish were exposed to normal operational routines in the farm. In these conditions, the shelters had no negative effects on fish growth performance or welfare. Further, from an operational perspective, no issues were found associated to the use of the shelters in the farm. The covers only caused minimal addition of workload related to daily cleaning procedures. Altogether, the results constitute a proof of concept for the use of shelters as a potential strategy for environmental enrichment in trout farming, but long-term studies will be necessary to explore the full potential of the use of shelters to promote the welfare of farmed trout. Furthermore, another dimension of structural enrichment, not considered here, relates to the materials the enrichment is made of, as plastic materials can contribute to the presence of microplastics or to the leaching of compounds of concern. Future work should focus on finding the best materials for adding structures for tank enrichment in fish rearing facilities.

## Figures and Tables

**Figure 1 animals-13-00268-f001:**
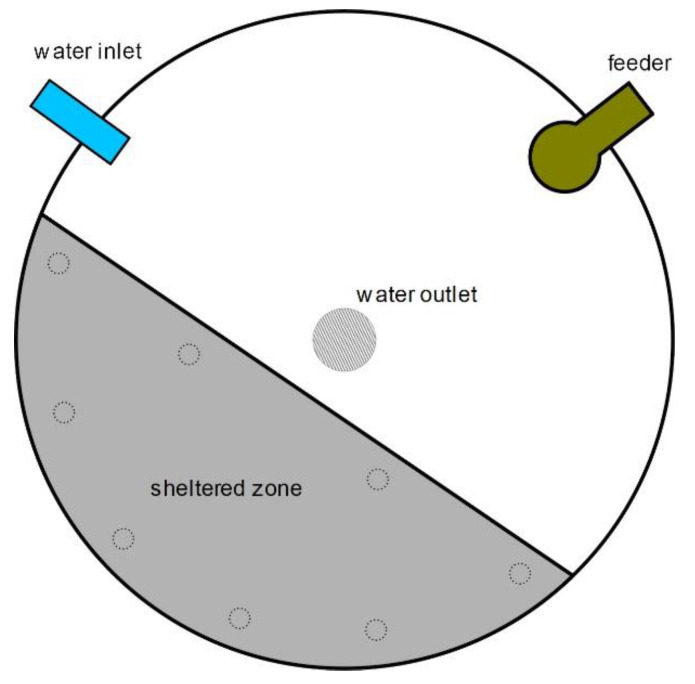
Scheme showing the positioning of the immersed covers in the sheltered tanks. Dash-line circles indicate the position of the legs of the shelter.

**Figure 2 animals-13-00268-f002:**
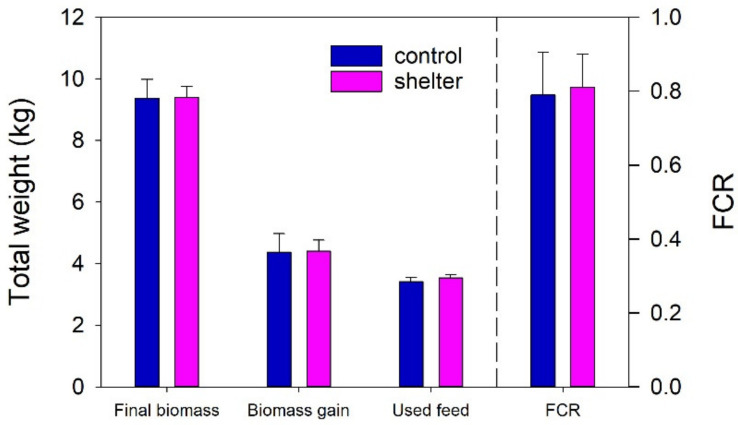
Growth performance (tank-based) of Control and Shelter rainbow trout (*Oncorhynchus mykiss*) fingerling groups during the 30 day experimental period. Data represent the mean and SD of *n* = 3 tanks per treatment. Student’s *t*-tests showed no treatment effect for Final biomass (*p* = 0.939), Biomass gain (*p* = 0.939), Used feed (*p* = 0.256) or feed conversion ratio (FCR, *p* = 0.821).

**Figure 3 animals-13-00268-f003:**
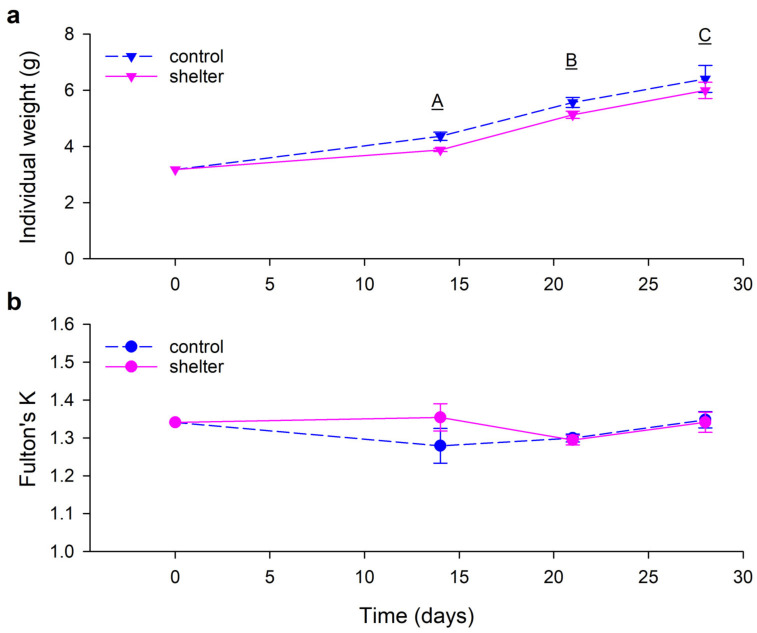
Individual mass (panel a) and Fulton’s condition factor (K, panel b) of rainbow trout (*Oncorhynchus mykiss*) fingerlings in Control and Shelter groups during the experimental period. Data represent the mean and SEM of *n* = 3 tanks per treatment. (**a**) Individual weight was affected by time (*p* < 0.001, different letters in the graph indicate significant differences among sampling times), but not by treatment (*p* = 0.067) or by the interaction time × treatment (*p* = 0.707). (**b**) Neither treatment (*p* = 0.104) nor time (*p* = 0.407) or their interaction (*p* = 0.424) affected K (two-way repeated measures ANOVA). No measurements were taken on day 7 due to technical failure of the weighing equipment.

**Figure 4 animals-13-00268-f004:**
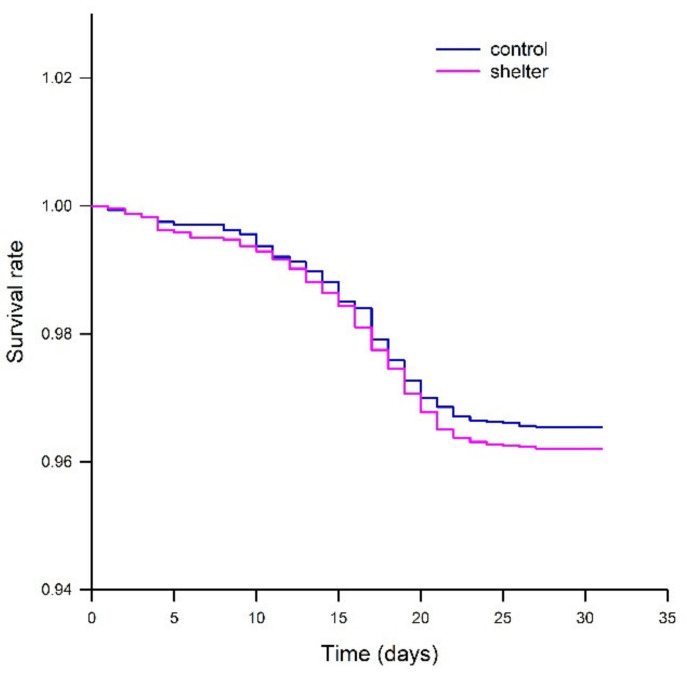
Kaplan-Meier survival curves for rainbow trout (*Oncorhynchus mykiss*) fingerlings in Control and Shelter groups during the experimental period. Mortality data from the different replicated tanks was merged for each treatment. Initial fish numbers were assumed equal for all tanks (1572 fish per tank). Treatment had no effects on survival (Log-rank, *p* = 0.382).

**Figure 5 animals-13-00268-f005:**
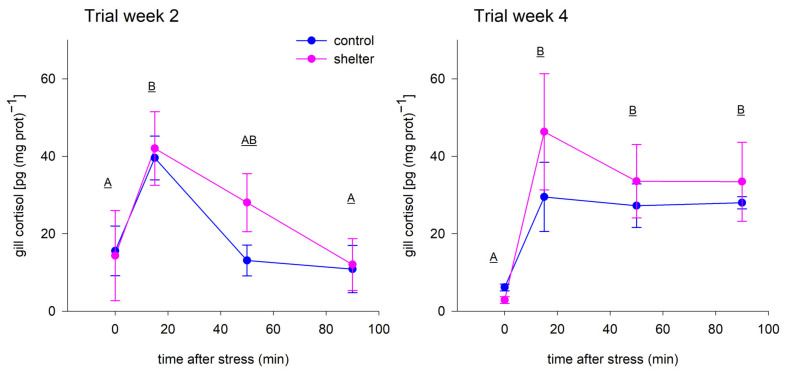
Gill levels of cortisol of rainbow trout (*Oncorhynchus mykiss*) fingerlings from Control and Shelter groups exposed to a standardized 2 min net-chasing stressor. Fish were exposed to the stressor on two occasions: after 2 weeks (left panel) and after 4 weeks (right panel) from the start of the experimental period. Data represent the mean and SEM of *n* = 3 tanks per treatment. Post-stress sampling time had an effect on cortisol levels on both occasions (two-way ANOVA, *p* = 0.013 and *p* < 0.001 for week 2 and week 4 trial, respectively); differences among sampling times are indicated by different letters. The presence of shelters did not affect cortisol levels in any of the trials (*p* = 0.624 and *p* = 0.521 for week 2 and week 4 trial, respectively). No significant time × treatment interactions were found (*p* = 0.660 and *p* = 0.543 for week 2 and week 4 trial, respectively).

**Figure 6 animals-13-00268-f006:**
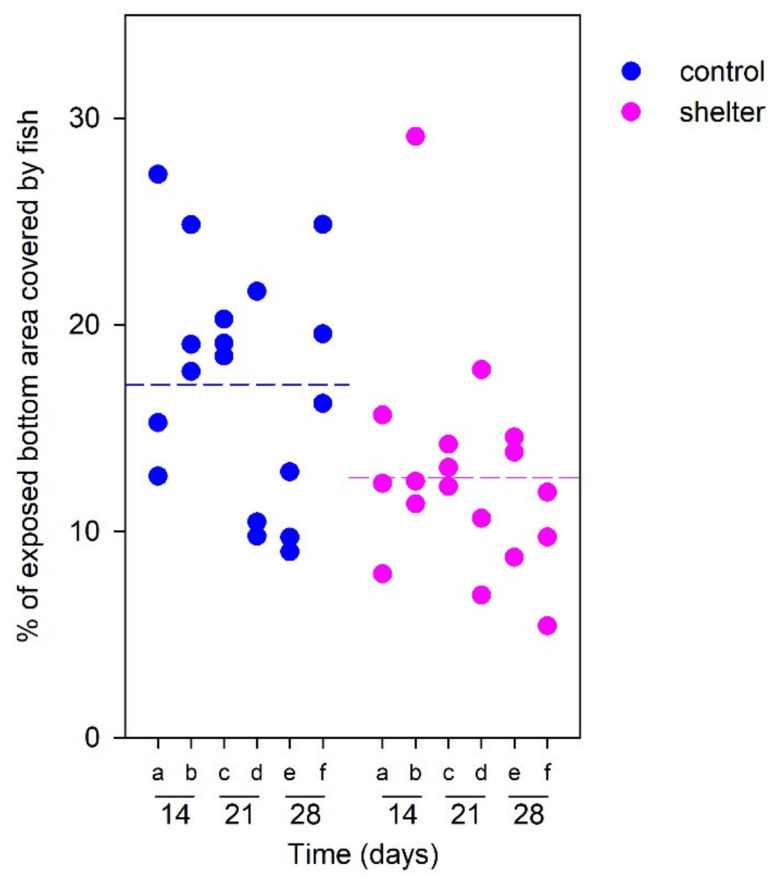
Spatial distribution of rainbow trout (*Oncorhynchus mykiss*) fingerlings in tanks with and without immersed cover. Data represent quantification of placement (percentage of exposed tank bottom area covered by fish in zenithal pictures), obtained in six different occasions: Events “a” to “f” correspond to zenithal pictures obtained in the morning or afternoon of a day during day 14 (a, b), day 21 (c, d), and day 28 (e, f) of the experimental period. High values indicate high presence of fish in the exposed (not-sheltered) part of the tank. Dotted lines indicate the mean values for both treatment groups. A significant effect of cover was found (*p* = 0.038, two-way repeated measures ANOVA), where exposed fish in tanks with cover were less densely distributed compared to fish in open tanks. No temporal effects were found after comparing sampling events (*p* = 0.117) and there was no interaction between both factors (*p* = 0.246).

**Figure 7 animals-13-00268-f007:**
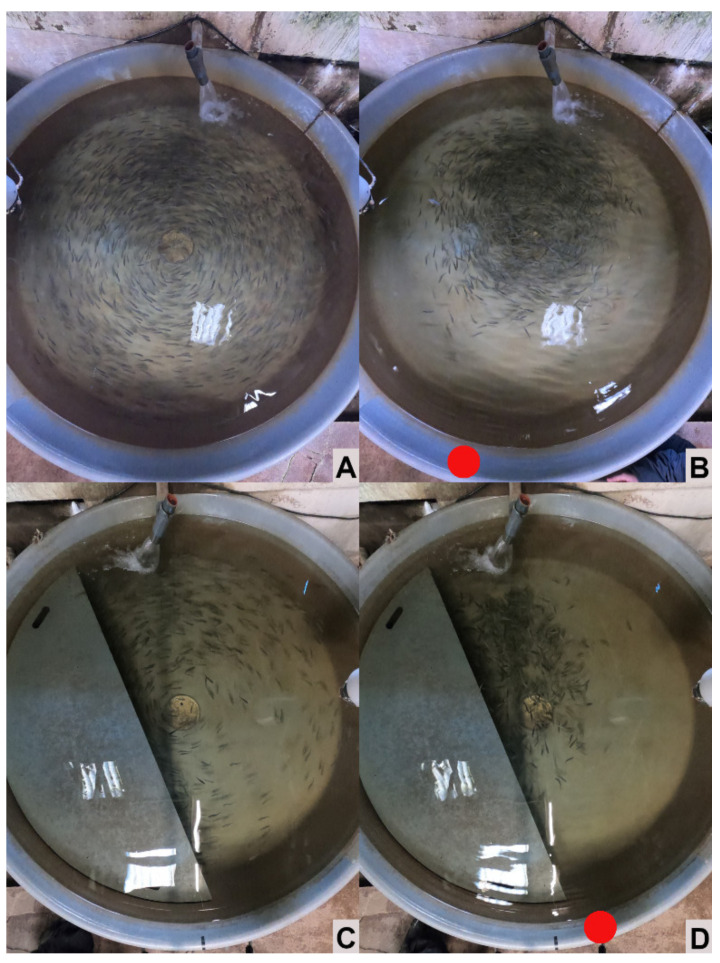
Representative zenithal pictures showing the usual swimming behavior of the rainbow trout (*Oncorhynchus mykiss*) fingerlings in the control (**A**,**B**) and sheltered tanks (**C**,**D**). Left panels (**A**,**C**) show fish swimming in undisturbed conditions, while panels (**B**,**D**) show the typical response of the fish just after disturbance (knocking on the tank wall): Fish tended to get away from the tank walls and group together, and this behavior appeared similar for both control and sheltered tanks. Red circles in (**B**,**D**) indicate the knocking spot. See the main text for more details.

## Data Availability

All data will be available upon request to the authors.

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
