# Peer review of "Environmental Enrichment for Rainbow Trout Fingerlings: A Case Study Using Shelters in an Organic Trout Farm"

_animals, 2023, doi:10.3390/ani13020268_

Round 1
Reviewer 1 Report
Properly planned, prepared and written manuscript. My few remarks are included in the text in the form of comments. To see them all, open the file in Acrobat Reader.

Author Response
Properly planned, prepared and written manuscript. My few remarks are included in the text in the form of comments. To see them all, open the file in Acrobat Reader.
Reply: Thank you for the very pertinent comments. All corrections have been made in the main text and figure captions as suggested. Modification made in the text are tracked, so they are easy to see in the revised manuscript.
Reviewer 2 Report
The research question is very interesting and relevant since enrichment in aquaculture is a very hot topic and can be quite controversial.
General comments
The experiment was well designed by the authors but its implementation at the farm showed some practical challenges and limitations (i.e the disease outbreak, the zenithal picture and the whole behavioral test protocol).
Regarding the behavioral test protocol, it is good that the authors present it as qualitative assessment, because of the technical issues encountered. The protocol used is well designed as well, the authors should have filmed (or taken more series of pictures over a longer period) to assess the acute response to the disturbance, to assess the immediate response to the fish, only using five images right after the disturbance doesn’t show the whole acute reaction it is just a snapshot. In addition, it is not said where the knocking on the tank happened, it could influence the movement of the fish, especially if done away from the sheltered area in the tanks with shelters. Was the knocking done always at the same spot in the tanks, for all tanks? The authors should show on the Fig 1 or Fig 7 where it has been done, if it was always carried out at the same spot, and it should also be described in the text.
I would suggest to also reduce the text describing this part of the experiment, and especially the part 2.7 (part of it actually belongs to the result/discussion) and 2.8 because of the limitations of the studies. For instance most of the 2.8 could be in a supplementary material.
It should be mentioned in the discussion as well about the potential impact that PVC (and plastics) enrichment could have on the fish health and welfare. Plastics and microplastics are also quite a hot topic, and there is a lot of research done at the moment on it, also in aquaculture. It should be mentioned as a potential concern for implementation/choice of enrichment material.
Specific comments
Introduction.
L43 and L49, please be more specific, how it could improve welfare and how it could be neutral or detrimental (give examples, i.e by decreasing water quality, increasing aggression)
L67, the first time you mention PVC, you should also say the full term, Polyvinyl chloride
L67, add the latin name of the trout species, and add ‘danish’ (or ‘in Denmark’)
L73, please be more specific, a clear preference, in term of?
L75-79, add the time frame of the experiment (including when the two stress trials were done)
L77, and also throughout the manuscript,L256, L356, and potentially elsewhere, please change ‘blood’ or ‘plasma’ by ‘gill’ cortisol to harmonize.
Material and methods
L102, L139, and potentially elsewhere, please harmonize, ‘s.e.m’, ‘SD’, change s.e.m to SEM?
L106. Why did you choose to cover the tanks with 1/3? Please mention your reasoning
L114, Fig 1. You should also show where the 8 legs are located (i.e make a Fig 1 b with a different angle, or use a 3D schematic, or just make 8 small circles to show the placement on the figure you have.
There is a potential experimental bias due to the placement of the sheltered zone, the food is given in the bright zone, far away from the feeder. It could have been nice to either place the shelter closer to the feeder of have several tanks with different location of the feeder. You should discuss it.
L118, which method/material were used to measure those?
L128. How long was the daily cleaning procedure?
L150. What is the brand/city/country of the manufacturer of the benzocaine
L161, L208 and elsewhere, you should harmonize (min, minutes)
L188, where were the knocks done on the tank? Please show it on Fig 1 or Fig 7
L215, L248 please add the reference of ImageJ and Sigmaplot(city/country)
L222, you should add a picture of what was the final result of the treatment to show the reader (add it in the supplementary material).
L236, please add the city/country of AquaOx and the brand/city/country of florfenicol
L245, what are you conclusions?
Results and discussion.
I think this part should be compartmented like the material and methods, for more clarity, a part on growth and survival (L275-330), a part on stress challenges/cortisol (L331-378) and a small part of behavior (L379-416).
L282, Fig 2, Fig 3, use ‘weight’ like in the other part of the text, instead of ‘mass’
Fig 3. It is quite confusing to see the individual weight and the K index on the same graph, consider splitting it into 3a and 3b. Also. Please calculate and add the Fulton’s K index for T0 as well, and mention again in the legend that there was no individual recording on week 1 for technical reasons
Fig 1, Fig 4, Fig 5 and Fig 6. In the legend, you should say clearly why color is which group
Fig 5. L361-362. There are no letters on the figure, are they missing? Or there are no differences? In case of no difference, delete this sentence. L356-365, you should reduce the font of the legend to match the other figures.
Fig 2, 4 and 5, please harmonize the X axis, use ‘Time (days)’ instead of week, days, and ‘a,b,c…’
L309-312, please rephrase.
I would also suggest to put the positive effect reference when you say ‘including positive [ref XX, ref YY], neutral [ref ZZ, UU] and negative [ref AA, BB]’ instead of listing all the references at the end of the sentence, so the readers knows which study had which effect.
L379-416. As mentioned earlier, I think this part is quite weak because of the design (i.e the challenges that you had, plus the fact that only taking pictures right after is just a snapshot and you cannot really conclude anything from it. Please reduce this part and discuss the limitation of your protocol and propose improvements.
L417-428. Here you can also discuss future perspective, i.e refining the behavioral protocol according the problems you have, plus discuss potential problem of using PVC as enrichment (i.e plastic/leaks)
Author Response
Please see the attached file. All responses to reviewer's comments are written in blue color font.

Round 2
Reviewer 2 Report
Thank you very much for addressing all my comments. I am happy with the detailed answers from the authors and the discussion.
I would suggest the manuscript to be accepted for publication.
Some final suggestions.
You can compartment figure 3 into a) and b) or use 'top'/'bottom' in its legend so it is more clear which part of the legend correspond to which graph (and eventutaly mention it in the text, fig 3a/fig 3b when needed).
Some references are missing doi number, I am not sure if it's the authors' responsibility or if the Animals editing team will fix it.
Kind regards